# Cannabidiol and Beta-Caryophyllene Combination Attenuates Diabetic Neuropathy by Inhibiting NLRP3 Inflammasome/NFκB through the AMPK/sirT3/Nrf2 Axis

**DOI:** 10.3390/biomedicines12071442

**Published:** 2024-06-28

**Authors:** Islauddin Khan, Sukhmandeep Kaur, Arun K. Rishi, Breana Boire, Mounika Aare, Mandip Singh

**Affiliations:** 1College of Pharmacy and Pharmaceutical Sciences, Florida A&M University, Tallahassee, FL 32307, USA; islauddinniper@gmail.com (I.K.); jsukhmand347@gmail.com (S.K.); breana1.boirie@famu.edu (B.B.); aaremounika111@gmail.com (M.A.); 2John D. Dingell Veterans Affairs Medical Center, Department of Oncology, Wayne State University School of Medicine, Detroit, MI 48201, USA; rishia@karmanos.org

**Keywords:** beta-caryophyllene, cannabidiol, AMPK, sirT3, NLRP3, mitochondrial biogenesis, autophagy

## Abstract

Background: In this study, we investigated in detail the role of cannabidiol (CBD), beta-caryophyllene (BC), or their combinations in diabetic peripheral neuropathy (DN). The key factors that contribute to DN include mitochondrial dysfunction, inflammation, and oxidative stress. Methods: Briefly, streptozotocin (STZ) (55 mg/kg) was injected intraperitoneally to induce DN in Sprague–Dawley rats, and we performed procedures involving Randall Sellito calipers, a Von Frey aesthesiometer, a hot plate, and cold plate methods to determine mechanical and thermal hyperalgesia in vivo. The blood flow to the nerves was assessed using a laser Doppler device. Schwann cells were exposed to high glucose (HG) at a dose of 30 mM to induce hyperglycemia and DCFDA, and JC1 and Mitosox staining were performed to determine mitochondrial membrane potential, reactive oxygen species, and mitochondrial superoxides in vitro. The rats were administered BC (30 mg/kg), CBD (15 mg/kg), or combination via i.p. injections, while Schwann cells were treated with 3.65 µM CBD, 75 µM BC, or combination to assess their role in DN amelioration. Results: Our results revealed that exposure to BC and CBD diminished HG-induced hyperglycemia in Schwann cells, in part by reducing mitochondrial membrane potential, reactive oxygen species, and mitochondrial superoxides. Furthermore, the BC and CBD combination treatment in vivo could prevent the deterioration of the mitochondrial quality control system by promoting autophagy and mitochondrial biogenesis while improving blood flow. CBD and BC treatments also reduced pain hypersensitivity to hyperalgesia and allodynia, with increased antioxidant and anti-inflammatory action in diabetic rats. These in vivo effects were attributed to significant upregulation of AMPK, sirT3, Nrf2, PINK1, PARKIN, LC3B, Beclin1, and TFAM functions, while downregulation of NLRP3 inflammasome, NFκB, COX2, and p62 activity was noted using Western blotting. Conclusions: the present study demonstrated that STZ and HG-induced oxidative and nitrosative stress play a crucial role in the pathogenesis of diabetic neuropathy. We find, for the first time, that a CBD and BC combination ameliorates DN by modulating the mitochondrial quality control system.

## 1. Introduction

A common microvascular complication of chronic diabetes, diabetic peripheral neuropathy (DN), impairs the blood supply to peripheral nerves, leading to loss in nerve conduction and other neuropathic manifestations [1,2]. The number of people suffering from diabetes is anticipated to rise from the current 536.6 million cases to 783.2 million by 2045 [3]. Looking at the world demographics, 8.4% of Chinese, 48.1% of Sri Lankans, 29.2% of Southeast Asians, 56.2% of Yemenis, 39.5% of Jordanians, 71.1% Nigerians, 16.6% of Ghanaians, and 29.5% of Ethiopian have DPN [4]. Furthermore, in Norh America, 47 percent of patients with diabetes have some form of neuropathy. Two drugs that are frequently prescribed for neuropathic pain are pregabalin and duloxetine, which function to provide symptomatic relief [5,6]. A general lack of treatment for DN and the incomplete understanding of its pathobiology require further research for the development of innovative and effective therapies to overcome DN symptoms.

According to recent studies, decreased biogenesis and mitochondrial function are the main factors driving DN progression [7,8]. The changes in the cellular NAD^+^/NADH ratio are mostly caused by a metabolic protein known as Sirtuin-1 [9,10]. By deacetylating transcription factors, Sirtuin-1 (SIRT1) stimulates downstream targets involved in the biogenesis of mitochondria and antioxidant defense [11,12]. SIRT1 signaling is essential for DN because it regulates mitochondrial function and antioxidant enzymes [12,13,14]. Nrf2 activation significantly improves mitochondrial function in both preclinical and clinical experiments [15,16]. Reactive oxygen species (ROS), which oxidize cellular proteins, are produced when hyperglycemia causes electron leakage from mitochondria [17,18]. This reduces antioxidant defense systems such as heme oxygenase-1, superoxide dismutase, and NADPH dehydrogenase quinone 1 [17,19]. Numerous studies have demonstrated that SIRT1 suppresses inflammatory responses by inhibiting the NLRP3 inflammasome in vascular endothelial cells [20,21]. Multiple investigations have also shown that SIRT1 activation is neuroprotective in DN, in part by enhancing mitochondrial bioenergetics and autophagy [22,23]. Additional studies have also demonstrated that AMPK and SIRT1 help to reduce DN by improving mitochondrial function through the PGC-1α and Nrf2 axes [24,25,26].

Animal models are utilized to replicate human clinical problems swiftly and accurately, since disease progression in animal models often follows a pattern similar that noted in human pathologies. As a result, evaluations of potential novel targets for their expected therapeutic benefits become straightforward and feasible [27,28,29,30]. Sprague–Dawley (SD) rats show a human-like genetic background, in addition to having clear genetic tractability [31]. The use of STZ in the generation of type 1 diabetes mellitus (T1DM)-induced DN has been validated by numerous published reports [32,33]. The peripheral nerves in these rats undergo pathological changes after the induction of hyperglycemia, including axonal atrophy and demyelination, which eventually leads to decreased sensory and motor nerve conduction [34,35,36]. Schwann cells have been shown to be sensitive to insulin and glucose, which are involved in diabetic peripheral neuropathy (DPN) pathogenesis. It has also been reported that high glucose causes Schwann cell apoptosis in the development of DN [37,38,39,40].

Cannabinoids, terpenoids, and flavonoids are just a few of the more than 500 unique compounds found in cannabis. Some studies have suggested that the components of the cannabis plant may cooperate to produce superior therapeutic results in pathologies such as hyperglycemia-induced DN [41,42]. However, thus far, only a few of these cannabinoids have had their benefits investigated. Furthermore, the components of the cannabis plant, when used in certain combinations, may produce superior therapeutic results, possibly due to the entourage effect [43,44,45,46,47].

The goal of this study was to ascertain whether the interaction between a terpene (β-caryophyllene, BC) and a minor cannabinoid (cannabidiol, CBD) has therapeutic potential for the treatment of hyperglycemia-induced DN. CBD is known to be an isomer of Δ9- tetrahydrocannabinol, the main psychoactive substance found in Cannabis sativa [48,49]. Cannabidiol pharmacology is intricate and is yet to be comprehensively investigated. It should be noted that CBD has a low affinity for cannabinoid receptors CB1 and CB2 and is thought to act on a number of different targets (e.g., 5-Hydroxytrypyamine (5HT1A)) instead [50]. β-caryophyllene (BC), a sesquiterpene that is additionally present in clove and black pepper, is one of the terpenes that are also most prevalent in cannabis. It was shown that BC is a naturally occurring, selective CB2 receptor agonist, with positive effects such as analgesia, antioxidant protection, anti-inflammation, and neuroprotection [41,51]. Therefore, there is a need to determine the beneficial and undesirable (negative) consequences of the medical use of cannabinoids, given the fast-changing legislation affecting access to cannabinoids, and to promote interest in the potential medical applications of cannabis (especially for DN) [47,52]. In particular, the combination of cannabinoid compounds could result in superior benefits compared with the benefits noted from their separate use. Here, we tested a hypothesis that a combination of a terpene (BC) and a minor cannabinoid (CBD) has superior therapeutic potential in the treatment of DN [45,46,53,54,55].

We conducted studies to determine the protective effects of the combination of BC and CBD on HG-induced Schwann cells neurotoxicity in vitro and STZ-induced DN in rats by performing behavioral, functional, and antioxidant analyses. Furthermore, ROS generation, changes in mitochondrial membrane potential, and mitochondrial superoxides were evaluated using Western blotting, immunohistochemistry, and intraepidermal nerve fiber (IENF) density to determine the impact of BC and CBD on mitochondrial dysfunction and mitochondrial biogenesis.

## 2. Material and Methods

### 2.1. Experimental Design

Human Schwann cells (obtained from ATCC) were cultivated at 37 °C in a humidified environment of 95% air and 5% CO_2_ and cultured in DMEM (5.5 mM glucose) enriched with 10% FBS, glutamine (2 mM), and streptomycin/penicillin (1%). Cell culture conditions with high glucose were created by adding 24.5 mM glucose to the solution, resulting in a final glucose concentration of 30 mM. In the cell culture system, the diabetic condition is defined as glucose levels that exceed 10 mM [56,57]. This study used a 30 mM glucose concentration that has been previously used to promote in vitro diabetic conditions in many cellular models of diabetes [58,59,60,61,62,63]. The Schwann cells were plated into a 6-well plate in DMEM medium containing 10% FBS. The treatment groups consisted of (a) normal control cells (NC), (b) high-glucose (HG)-treated cells (30 mM), (c) HG-induced cells treated with 75 µM of BC (HG+BC), (d) HG-induced cells treated with 8 µM of CBD (HG+CBD), and (e) HG-induced cells treated with 75 and 3.64 µM of BC and CBD (HG+BC+CBD), respectively. After 24 h of treatment, various biochemical and molecular parameters were measured.

### 2.2. Cell Viability Determination

With a density of roughly 5000–8000 cells per well, the Schwann cells were plated in 96-well plates with DMEM medium and maintained for 24 h. After 24 h of incubation, the cells were exposed to various concentrations of CBD (100–3.125 µM), BC (500–15.625 µM), and BC+CBD (8–0.25 µM), respectively. Cell viability was determined using a MTT assay. Briefly, 100 µL of MTT (5 mg/10 mL) was added to the plates after 24 h of treatment, and the plates were then maintained in an incubator at 37 °C for 4 h. The media was removed, and 100 µL of DMSO was added to the 96-well plates for 10 min. The absorbance value was recorded at 570 nm using a TECAN infinite M200 plate reader [64].

### 2.3. DCFDA Staining to Measure Intracellular ROS

Schwann cells were seeded at a density of 5 × 10^4^ cells per well in a six-well plate and treated with HG (30 mM) as well as with BC (75 µM), CBD (8 µM), BC+CBD (75 µM + 3.64 µM) for 12 h. After 12 h, 10 µM of DCFDA dye (Invitrogen by Thermo Fisher Scientific, Eugene, OR, USA) were added to each well, and the cells were incubated for an additional 30 min, followed by PBS washing. The cells were then imaged using an OLYMPUS IX73 microscope, Tokyo, Japan. The images were analyzed using NIH ImageJ software (version 1.54) [65,66,67].

### 2.4. MitoSOX Staining to Measure Mitochondrial Superoxides

Schwann cells were seeded in 6-well plates as above and treated with HG (30 mM) as well as BC (75 µM), CBD (8 µM), BC+CBD (75 µM + 3.64 µM) for 12 h to measure the amount of mitochondrial superoxide anions produced. After a treatment period of 12 h, 5 µM of MitoSOX dye (Cat no. M36008; Thermo Fisher Scientific) was added. The cells were incubated for 15 min, followed by PBS washing. Fluorescence images were captured using an OLYMPUS IX73 microscope. The images were analyzed using ImageJ software [68,69].

### 2.5. JC1 Staining for Mitochondrial Membrane Potential

Hyperglycemia was induced by exposing the Schwann cells to HG (30 mM), followed by treatments with BC (75 µM), CBD (8 µM), and BC+CBD (75 µM + 3.64 µM) for 12 h. After 12 h, the cells were incubated for 15 min with 5 µM of JC-1 dye (Cat no. T3168; Thermo Fisher Scientific). After three PBS washes, the cells were photographed using an OLYMPUS IX73 microscope. The images were analyzed using ImageJ software [68,70,71].

### 2.6. Western Blotting

Schwann cells were seeded at a concentration of 2 × 10^6^ in 6-well plates with DMEM media. Hyperglycemia was induced by HG (30 mM), followed by BC (75 µM), CBD (8 µM), and BC+CBD (75 µM + 3.64 µM) treatment for 24 h. After performing a phosphate buffer wash, the cell lysate was prepared using RIPA buffer with a phosphatase inhibitor and protease inhibitor (1:100) and incubated for 30 min on ice. The supernatant was then collected by centrifuging the lysates at 12,000 rpm for 15 min. Bradford reagent was used to estimate the amount of protein in the samples. A total of 40–60 µg of equivalent protein was added to the loading buffer, heated at 98 °C for 10 min, and then electrophoresed on SDS–PAGE. After the electrophoresis, the proteins were transferred to nitrocellulose membranes, and the membranes incubated in 5% BSA in TBST. The membranes were then incubated with primary antibodies prepared at dilutions of 1:1000 to 1:3000 in the TBST at 4 °C overnight on a shaker. Various primary antibodies were purchased either from Cell Signaling Technology or Santa Cruz biotechnology. These antibodies included AMPK (Cat no. 2532S), SIRT1 (Cat no. 9475S), Sirt3 (Cat no. 5490S), PGC-1α (Cat no. 2178S), NRF1 (Cat no. 46743S), PINK1 (Cat no. SC517353), PARKIN (Cat no. SC32282), p62 (Cat no. 5114S), TFAM (Cat no. 8076S), SOD2 (Cat no. SC30080), Nrf2 (Cat no. 12721S), KEAP1 (Cat no. 8047S), HO-1 (Cat no. SC390991), NQO1 (Cat no. SC376023), pMTOR (Cat no. 2971S), NLRP3 (Cat no. 15101S), ASC (Cat no. 67824S), Caspase-1 (Cat no. 3866S), IL-1 β (Cat no. 12242S), IL-18 (Cat no. 54943S), PHB2 (Cat no. 14085S), BAX (Cat no. 2772S), BCL2 (Cat no. 3498S), COX2 (Cat no. 12282S), Phospho-NF-κB p65 (Ser536) (93H1) (Cat no. 3033S), FOXO3a (Cat no. 2497S), Beclin1 (Cat no. 3495S), atg3 (Cat no. 3414S), atg7 (Cat no. 8558S), LC3B (Cat no. 2775S) (CST, Danvers, MA, USA), and β-actin (Cat no. 4970L). After incubation with the primary antibodies, the membranes were washed three times for 10 min each. Finally, secondary antibodies tagged with HRP anti-rabbit (Cat no. 7074S), or anti-mouse (Cat no. 7076S) were added onto the membrane and kept at room temperature for 1.30 h on a shaker followed by washing with TBST. The signals for the presence of the respective proteins on the NC membrane were captured using a ChemiDoc^TM^ XRS+ Imaging system (BIO-RAD Molecular Imager), and the relative band intensities were quantified using densitometry ImageJ software [72,73,74,75,76].

### 2.7. In Vivo Experiments

A total of 32 healthy male Sprague–Dawley rats, aged three months and weighing 200–280 g, were kept in plastic cages with a 12 h light/dark cycle and were supplied a regular diet and water as needed. The American Association for Accreditation of Laboratory Animal Care (AAALAC) has given FAMU approval for the facilities where the animals were kept and cared for in accordance with strict pathogen-free standards. The animals were given a week of acclimatization prior to the commencement of studies following Florida A&M University’s Institutional Animal Care and Use Committee (IACUC) standards (protocol number 021–04, dated 05/10/2021) and Animal Research Reporting of in vivo studies (ARRIVE) guidelines. Efforts were made to reduce the number of animals used and their distress [77].

### 2.8. Induction of Diabetic Neuropathy and the Experimental Design

SD male rats were selected and fasted for 2 h. STZ at 55 mg/kg, dissolved in the citrate buffer, was given through the i.p. route to induce diabetes. Rats with their plasma glucose levels higher than 250 mg/dl were taken as diabetic for this study [32,33,78,79]. The animals with diabetes were randomized into groups consisting of a diabetic control animal group (STZ, *n* = 8); diabetic rats treated with 30 mg/kg BC, i.p. (STZ+BC30, *n* = 6) and 15 mg/kg CBD, i.p. (STZ+CBD15, *n* = 6); and diabetic rats treated with 30 mg/kg BC+15 mg/kg CBD, i.p. (STZ+BC30+CBD15, *n* = 6) after the 5th week of DN induction. The dosing was performed thrice a week for three weeks. Age-matched control rats were included as normal control animals (NC, *n* = 6). The treatment scheme is depicted in Figure 1. After the last dose of drug administration, behavioral and functional parameters were evaluated, followed by animal sacrifice with sciatic nerve, spinal cord, DRG and hind paw collection. The tissues were stored at −80 °C for Western blotting and in formalin for histology. Both CBD and BC were dissolved in a vehicle comprised of 5% DMSO, 5% ethanol, 5% Tween 80, and 85% saline [45].

### 2.9. Behavioral Parameters

#### Mechanical and Thermal Hyperalgesia

A cutoff time of 15 s was used to record the time duration to flick the rat’s tail during hot plate tests at a temp of 45 °C and cold plate tests at a temp of 10 °C, with a 30 s interval between each reading. At least three readings were taken to calculate the paw withdrawal time in seconds [33]. The von Frey aesthesiometer test and Randall Sellito calipers (IITC life sciences, Woodland Hills, CA, USA) were used to determine mechanical hyperalgesia. The animal paw withdrawal threshold in grams was determined based on the force at which it was observed. A minimum of 5 readings were taken, with a 5 min gap between each reading [69].

### 2.10. Nerve Functional Studies

#### Nerve Blood Flow

According to our earlier findings, a laser doppler oxymeter (Moor Instruments, Devon, UK) was used to measure the blood flow in the animals’ sciatic nerves. The sciatic nerve of the sedated animals was exposed on the left flank, and by utilizing Moor instrument software Devon, UK 6th generation. the flux was monitored for 10 min. The arbitrary perfusion units for the animals were recorded [80].

### 2.11. Western Blotting

The sciatic nerve was taken from −80 °C, and lysates were prepared using tissue extraction reagent (TPER, Sigma Aldrich, St. Louis, MO, USA) containing protease and phosphatase inhibitors. The remainder of the procedure was followed as mentioned above in the Schwann cell line WB experiments.

### 2.12. Immunohistochemistry (IHC)

Briefly, deparaffinized rehydrated 5 µm microsections of sciatic nerves were heated in citrate buffer at pH 6.0 for antigen retrieval. The sections were treated with 3% H_2_O_2_ to prevent endogenous peroxidase activity, followed by a 60 min blocking step with 3% BSA in PBS. The nerve microsections were then incubated with a SOD2 antibody at a 1:200 dilution (Cat no. SC30080), an LC3B antibody at a dilution of 1:400 (Cat no. 2775S), or an NF-κB p65 (D14E12) antibody at a 1:200 dilution (Cat no. 8242S) (CST, USA) for two hours at room temperature. The sections were then washed with tris-buffered saline (pH 7.4) before being incubated for 30 min at room temperature with anti-rabbit secondary antibodies (1:200) dilution each; CST, USA). Chromogenic staining was performed using the VECTASTAIN^®^ Elite ABC Reagent kit (Vector Labs, Newark, CA, USA) according to the manufacturer’s instructions, and the emergence of a brown color was noted. The sections were counterstained with hematoxylin, dehydrated, and mounted with DPX. To ascertain the immunopositivity, each segment was examined using an OLYMPUS IX73 microscope. Image analysis was performed using ImageJ software [80].

### 2.13. Intra-Epidermal Nerve Fiber Density in the Hind Paw of Diabetic Rats

The diabetic rat hind paws were sliced into 8 µm sections and treated in accordance with the above IHC procedure. The nerve fibers were marked with the PGP 9.5 antibody (Invitrogen, Thermo Fisher Scientific; Cat no. 381000) at a 1:200 dilution. Photographs were captured at a 20× magnification, and the numbers of nerve fibers to the epidermal junction were calculated in relation to the length of the tissue under examination. In each group, at least five sections were counted, and the average number of nerve fibers/mm was derived. The images were analyzed using ImageJ software [68,80].

### 2.14. Statistical Analysis

The data were expressed as mean ± standard error of the mean (SEM) and were compared using a one way analysis of variance (ANOVA) using Graph Pad Prism followed by Benferroni’s multiple comparisons post hoc test. *p* < 0.05 was considered statistically significant.

## 3. Results

### 3.1. BC, CBD, and Their Combined Effect on Cell Viability

The MTT results revealed that the doses of BC, CBD, and BC+CBD at 75 µM, 8 µM, and 3.65 µM, respectively, were suitable to evaluate their protective potential in high-glucose-exposed Schwann cells (Figure 2).

### 3.2. Effects of BC, CBD, and Their Combination on the Generation of ROS, Mitochondrial Superoxide, and the Mitochondrial Potential in High-Glucose-Exposed Schwann Cells

DCFDA and Mito Sox labelling were used to evaluate the ROS and the mitochondrial superoxide levels in Schwann cells. As compared to cells in a normoglycemic environment, we observed that hyperglycemia induced a significant (*p* < 0.001) rise in ROS and the mitochondrial superoxide (Figure 3A–D). BC and CBD treatments at 75 and 3.65 µM, respectively, resulted in significant (*p* < 0.001) reductions in the ROS and superoxide levels compared to the HG treatment group (Figure 3A,D).

JC1 labelling showed two separate cellular populations: a green fluorescence that indicated JC1 in its monomeric state and an orange fluorescence that signaled the formation of JC1 aggregates, indicating negatively polarized mitochondria. When compared to the high-glucose conditions, BC, and CBD treatment at 75 and 3.65 µM doses, respectively, it restored a negative MMP (mitochondrial membrane potential) (*p* < 0.001) (Figure 3E,F).

### 3.3. BC, CBD, and Their Combination Effects on Mitochondrial Biogenesis and Antioxidant Effects in HG-Induced Schwann Cells

AMPK, SIRT1, and PGC-1α play significant roles in mitochondrial quality functions, especially in the mitochondriogenesis and autophagy. Figure 4A shows expression of AMPK, SIRT1, PGC-1α, Nrf2, HO-1, SOD2, and NQO1 in Schwann cells that were either untreated or treated with HG, HG+BC, HG+CBD, and HG+BC+CBD. The mitochondrial protein expression was significantly decreased for AMPK (*p* < 0.001), SIRT1 (*p* < 0.001), PGC-1α (*p* < 0.001), Nrf2 (*p* < 0.001), HO-1 (*p* < 0.001), SOD2 (*p* < 0.001), and NQO1 (*p* < 0.001) in HG-induced Schwann cells when compared to the untreated cells (Figure 4B). The quantitation analyses in Figure 4B also showed that the BC and CBD combination treatment significantly increased the AMPK (*p* < 0.01), SIRT1 (*p* < 0.001), PGC-1α (*p* < 0.001), HO-1 (*p* < 0.001), SOD2 (*p* < 0.001), and NQO1 (*p* < 0.001) expression levels relative to their HG-induced counterparts. The results suggest that the BC and CBD combination provided a protective mitochondriogenesis by promoting increased expression of AMPK, PGC-1α, and SIRT1 in HG-induced cells that likely function to restore normal mitochondrial function (Figure 4A,B).

### 3.4. Effects of BC, CBD, and Their Combination on Neuroinflammation in HG-Induced Schwann Cells

NLRP3, ASC, IL-1β, IL18, pNFkB, and COX2 are well-known pro-inflammatory signaling molecules. Figure 5A shows the expression of NLRP3, ASC, IL-1β, IL18, pNFkB, and COX2 in Schwann cells that were either untreated or treated with HG, HG+BC, HG+CBD, or HG+BC+CBD. The expression and/or phosphorylation (activation) of NLRP3 (*p* < 0.001), ASC (*p* < 0.001), IL-1β (*p* < 0.001), and IL18 (*p* < 0.001) was significantly elevated in HG-treated Schwann cells compared to the normal, untreated cells (Figure 5B). The quantitation analyses in Figure 5B also showed that the BC and CBD combination significantly decreased NLRP3 (*p* < 0.001), ASC (*p* < 0.001), IL-1β (*p* < 0.001), and IL18 (*p* < 0.001) expression relative to their HG-induced counterparts, indicating that HG promotes inflammasome activation that could enhance neuroinflammation, which was reduced by the BC and CBD combination treatment. Additionally, pNFkB (*p* < 0.001) and COX2 (*p* < 0.001) were also significantly elevated in the HG-treated Schwann cells, while there was a significant reduction in the levels of pNFkB (*p* < 0.001) and COX2 (*p* < 0.001) in the cells treated with HG+BC, HG+CBD, or HG+BC+CBD relative to the HG-treated cells (Figure 5B). Together, these data suggest that the BC and CBD combination had an anti-inflammatory action in the HG-treated Schwann cells (Figure 5A,B).

### 3.5. Effects of BC and CBD Combination on Mitochondrial Quality Control and Autophagy in High-Glucose-Induced Schwann Cells

The expressions of the mitochondrial proteins sirT3, PHB2, PINK1, and PARKIN played roles in mitochondrial quality control and the regulation of autophagy. Figure 6A shows expression of sirT3, PHB2, PINK1, PARKIN, pMTOR, p62, and LC3B in Schwann cells that were either untreated or treated with HG, HG+BC, HG+CBD, or HG+BC+CBD. The quantitation analyses in Figure 6B further show that the Schwann cells, when exposed to HG (30 mM), exhibited significant decreases in SirT3 (*p* < 0.001), PHB2 (*p* < 0.001), PINK1 (*p* < 0.001), PARKIN *(p* < 0.001), and LC3B (p<0.001), while expression of phospho-mTOR (*p* < 0.001), p62 (*p* < 0.001) was significantly increased when compared with normal, untreated cells, indicating that the HG treatment likely impacted mitochondrial quality and promoted autophagy. The CBD and HG+CBD+BC treatment significantly enhanced sirT3 (*p* < 0.001), PHB2 (*p* < 0.001), PINK1 (*p* < 0.001), and PARKIN (*p* < 0.001) expression relative to their respective levels in HG-treated cells, suggesting a positive role of these proteins in maintaining mitochondrial quality control (Figure 6A,B). Additionally, the expression of phospho-mTOR (*p* < 0.001), p62 (*p* < 0.001) significantly increased, and LC3B (*p* < 0.001) expression significantly decreased in the HG-treated Schwann cells compared to normal Schwann cells, while the BC and CBD combination caused significant decreases in phospho-mTOR (*p* < 0.001), p62 (*p* < 0.001) and an increase in LC3B (*p* < 0.001) expression, suggesting that the BC and CBD combination played a significant protective role in mitochondrial autophagy relative to the HG treatments.

### 3.6. Effects of BC, CBD, and Their Combination on Nerve Function in SD Rats

The effects of BC, CBD, or their combination were investigated in the SD rat model as described in the Methods section. The animals were administered STZ to induce diabetes. The diabetic animals showed significantly reduced paw withdrawal latency responses in the cold plate (*p* < 0.001) (Figure 7A), hot plate (*p* < 0.001) (Figure 7B), and Hargreaves tests (*p* < 0.001) (Figure 7C). A similar reduction in such responses was also noted based on tbe mechanical Randall–Selitto (*p* < 0.001) (Figure 8B) and Von Frey (*p* < 0.001) (Figure 8A) studies. The BC and CBD treatments at doses of 30 and 15 mg/kg, respectively, for 3 weeks or a combination of both the agents significantly improved the paw withdrawal latency responses in the thermal cold plate (*p* < 0.001) (Figure 7A), hot plate (*p* < 0.001) (Figure 7B), and Hargreaves tests (*p* < 0.001) (Figure 7C) as well as the mechanical hyperalgesia, Randall–Selitto (*p* < 0.001) (Figure 8B),and von Frey (*p* < 0.001) (Figure 8A) readings in the diabetic rats. The STZ-induced diabetes over the 8-week period also impaired the nerve blood flow significantly (*p* < 0.001) (Figure 8C) compared to the normal control rats. The three-week treatment with BC and CBD (*p* < 0.001), significantly restored the nerve blood flow (*p* < 0.001) impairments associated with DN (Figure 8C).

### 3.7. Effects of BC, CBD, and Their Combination on Mitochondrial Biogenesis in Diabetic SD Rats

Diabetic animals were treated with BC, CBD, or a combination as noted above. Following the treatments, the sciatic nerves were analyzed first using immunohistochemistry for levels of SDO2 (Figure 9A). The diabetic SD rat sciatic nerve showed a visible decline in SOD2 staining when compared with the control group rats. The BC- and CBD-treated rats showed increased SOD2, as noted by brown staining in Figure 9A. The histogram in Figure 9B shows the relative levels of SOD2 in the stained sciatic nerves in Figure 9A. In the SD diabetic rat sciatic nerves, the SOD2 (*p* < 0.001) protein declined significantly, while the treatments with BC, CBD, or a combination resulted in a significant restoration of the SOD2 levels in the sciatic nerves (9B). Similar to our observations in the Schwann cell culture assays, Western blot analyses of the sciatic nerves in the diabetic rats also showed significant decreases in AMPK (*p* < 0.001) and SIRT1 (*p* < 0.001) expression when compared to their levels in the sciatic nerves of the normal control group animals. The decline in the expression of SIRT1 resulted in significantly decreased expression of PGC-1α *(p* < 0.001) in the sciatic nerves of the diabetic SD rats relative to the PGC-1α levels in the sciatic nerves of the normal control group rats (Figure 9C,D). Upon BC and CBD combination treatments, there was a significant increase in the levels of AMPK (*p* < 0.001), SIRT1 (*p* < 0.01), and PGC-1α (*p* < 0.01) in the sciatic nerves of the diabetic SD rats relative to the levels of these proteins in the sciatic nerves of their untreated counterparts (Figure 9C,D). As also noted in the in vitro studies, the STZ-treated diabetic rats had reduced levels of NRF1 (*p* < 0.001), TFAM (*p* < 0.001), SOD2 (*p* < 0.01), NQO1 (*p* < 0.001), and HO1 (*p* < 0.001) proteins compared to their respective levels in the control group rats. The BC and CBD treatment with 30 and 15 mg/kg body weight dose, respectively, resulted in significantly increased levels of NRF1 (*p* < 0.001), TFAM (*p* < 0.001), SOD2 (*p* < 0.01), NQO1 (*p* < 0.001), and HO1 (*p* < 0.001) proteins relative to their levels in the sciatic nerves of the diabetic animals (Figure 9C,D).

### 3.8. Effects of BC, CBD, and Their Combination on Inflammasome and Nrf2-Linked Antioxidant Effects in Diabetic SD Rats

The sciatic nerves from the control, untreated, and diabetic animals, as well as the diabetic animals that were treated with BC, CBD, or a combination, were analyzed first using immunohistochemistry to determine the levels of pNFkB as above (Figure 10A). The diabetic SD rat sciatic nerves showed visible increases in pNFkB staining when compared with the pNFkB staining in the control group rats. The BC- and CBD-treated rats showed decreases in pNFkB levels, as noted by brown staining in Figure 10A. The histogram in Figure 10B shows the relative levels of pNFkB in the stained sciatic nerves in Figure 10A. In the SD diabetic rat sciatic nerves, the pNFkB (*p* < 0.001) protein increased significantly, while treatments with BC, CBD, or a combination resulted in significant declines in pNFkB levels in the sciatic nerves (Figure 10B). Similar to our data in the Schwann cell culture assays, Western blot analyses of the sciatic nerves in the diabetic rats also showed significant increases in the pro-inflammatory genes NLRP3 (*p* < 0.001), IL18 (*p* < 0.001), COX2 (*p* < 0.001), pNFkB (*p* < 0.001), and Keap1 (*p* < 0.001), while decreases in the levels of Nrf2 (*p* < 0.001) and FOXO3a (*p* < 0.001) occurred relative to their respective levels in the sciatic nerves of the untreated control animals (Figure 10C,D). The BC and CBD at a dose of 30 and 15 mg/kg, respectively, for 3 weeks significantly decreased the levels of NLRP3 (*p* < 0.001), IL18 (*p* < 0.001), COX2 (*p* < 0.01), pNFkB (*p* < 0.05), and Keap1 (*p* < 0.001) while inducing expression of Nrf2 (*p* < 0.01), FOXO3a (*p* < 0.001) relative to their respective levels in the sciatic nerves of the diabetic animals (Figure 10C,D). Collectively, our data in Figure 10 suggest that the BC and CBD treatments lowered the inflammatory responses while promoting antioxidant signaling in the sciatic nerves of the diabetic animals.

### 3.9. Effects of BC, CBD, and Their Combination on Autophagy in the Diabetic SD Rats

Several mitochondrial proteins play important roles in mitochondrial quality control and the regulation of autophagy. On the basis of our in vitro studies in Schwann cells, we determined whether the induction of diabetes and subsequent interventions via the administration of BC and CBD altered levels of autophagy-regulating proteins in the sciatic nerves of the SD rats. First, the sciatic nerves from the control, untreated, diabetic animals, and the diabetic animals that were treated with BC, CBD, or a combination were analyzed using immunohistochemistry for levels of LC3B as above (Figure 11A). The diabetic SD rat sciatic nerve showed a visible decline in LC3B staining when compared with the LC3B staining in the sciatic nerve from the control group rats. The BC- and CBD-treated rats showed elevated levels of LC3B in their sciatic nerves relative to the LC3B levels in the sciatic nerves from the diabetic SD rats (Figure 11A). Western blot analyses of the proteins from the peripheral sciatic nerves of the diabetic rats also showed a significant decrease in the expression levels of the sirT3 (*p* < 0.001), PINK1 (*p* < 0.001), PARKIN (*p* < 0.001), LC3B (*p* < 0.001), Beclin1 (*p* < 0.001), atg3 (*p* < 0.001), and atg7 (*p* < 0.001) proteins, while there was a significant increase in the expression of p62 (*p* < 0.001) when compared to the levels of the respective proteins in the peripheral sciatic nerves of the control group rats (Figure 11C,D). BC and CBD at doses of 30 and 15 mg/kg, respectively, for 3 weeks significantly increased the sirT3 (*p* < 0.001), PINK1 (*p* < 0.001), PARKIN (*p* < 0.01), LC3B (*p* < 0.01), Beclin1 (*p* < 0.001), atg3 (*p* < 0.001), and atg7 (*p* < 0.001) expression levels, while there was a significant decrease in the expression of the p62 (*p* < 0.05) protein relative to its respective levels in the sciatic nerves of the diabetic animals (Figure 11C,D). Additionally, the level of the apoptotic marker BCL2 (*p* < 0.001) was significantly reduced, while BAX (*p* < 0.001) was significantly increased in the sciatic nerves of the diabetic rats compared to its levels in the sciatic nerves of the control group rats. BC and CBD treatments caused significant increases and decreases in the BCL2 (*p* < 0.001) and BAX (*p* < 0.001) protein levels, respectively, suggesting that the BC and CBD combination interventions served to inhibit apoptosis in DN (Figure 11C,D). The data in Figure 11 further support our hypothesis that BC and CBD play significant roles in the treatment of diabetic neuropathy, in part through the reduction of autophagy and nerve apoptosis.

### 3.10. Effects of BC, CBD, and Their Combination on the Loss of Intraepidermal Nerve Fiber (IENF) in the STZ-Induced Neuropathic SD Rats

As DN is known to cause degeneration of IENFs, we conducted histological staining of the nerve fibers obtained from the control, untreated, diabetic animals, and the diabetic animals that were treated with BC, CBD, or their combination. The diabetic SD rats showed a decline in IENF staining when compared with the IENF levels noted in the control group rats (Figure 12A). The BC- and CBD-treated rats showed elevated levels of IENFs relative to those noted in the diabetic SD rats (Figure 12A). The quantitation analyses in Figure 12B further demonstrated that the DN rats had significant IENF degeneration (*p* < 0.001) compared to the IENF in the control group rats, while the BC and CBD combination treatments resulted in significant restoration (*p* < 0.001) of IENF levels relative to those noted in their diabetic counterparts (Figure 12A).

## 4. Discussion

Our present study focused on the evaluation of the antioxidant, anti-inflammatory, and neuroprotective potential of BC and CBD in HG-promoted toxicity and STZ-induced diabetic neuropathy through alterations in AMPK and SIRT1 activation, which may also be connected to altered, PGC-1α-mediated, mitochondrial biogenesis, PARKIN–PINK1–p62-mediated autophagy, antioxidant effects through Nrf2 overactivation, and anti-inflammatory effects through the NFkB and NLRP3 inflammasome. Though many studies have reported the role of CBD and BC in neuropathy, this is the first study to evaluate the role of their combination in diabetic peripheral neuropathy [81,82,83].

Increased amounts of ROS, mitochondrial superoxides, and mitochondrial membrane depolarization of mitochondrial dysfunction were observed in the Schwann cells after high glucose exposure. Reduced superoxide generation following the BC and CBD therapy prevented high-glucose-dependent ROS formation, superoxide generation, and mitochondrial membrane depolarization. These effects of BC and CBD have demonstrated their mito-protective properties, thus providing a rationale for further assessment of the molecular underpinnings. These results also unambiguously imply that BC and CBD may protect against high-glucose-induced neurotoxicity by limiting cellular homeostasis and balancing oxidative stress and mitochondrial function [84]. A study by Hashish, H.M. and Ni, B. et al. [41] found that BC and CBD treatment prevented HG-induced mitochondrial dysfunction and ROS production, respectively, by impacting AMPK- and SIRT1-induced mitochondrial biogenesis and Nrf2-mediated antioxidant signaling [41,85]. These observations demonstrate that mitochondrial biogenesis deficits play a critical role in the pathophysiology of DN, and activation of AMPK and SIRT1 function restores this condition by promoting mitochondrial biogenesis. These findings are consistent with other studies that demonstrate that the activation of SIRT1 alleviates oxidative damage and restores mitochondrial biogenesis [33,69,86,87]. The current findings point to the involvement of autophagy and mitochondrial biogenesis in the pathophysiology of DN triggered by Type 1 DM and provide a proof-of-concept rationale for the use of CBD and BC as pharmacological moderators of mitochondrial dysfunction through SIRT1 and AMPK [85]. The characteristics of STZ-induced DN in rats include lower nerve conduction and blood supply to the peripheral nerves, as well as increased pain hypersensitivity to hyperalgesia and allodynia. In diabetic rats, abnormal nerve conduction in the peripheral nerve was connected to a variety of abnormal sensations, such as tingling and discomfort [88]. The advancement of peripheral neuropathy in rats was confirmed by the development of tactile allodynia, mechanical hyperalgesia, and lower NBF levels in rats with chronic hyperglycaemia. In this context, the study by Wang G et al. demonstrated that treatment with cannabidiol induces autophagy and improves neuronal health [89].

In diabetic rats, decreased AMPK Thr 172 phosphorylation impairs the development of autophagosomes and mitochondrial biogenesis. Our experiments also provided evidence that, in hyperglycemic settings, AMPK activation increases PGC-1α-mediated mitochondrial biogenesis and reduces the activation of NFκB-induced neuroinflammation. It is also known that NFκB-mediated neuroinflammation affects the vascular impairments seen in DN by inducing COX-2. BC and CBD combination treatments increased peripheral blood flow to the neurons of diabetic mice, which may have been due to AMPK activation of NFκB inhibition [32,90,91,92]. By enhancing mitochondrial activity, SIRT1 activation has also been associated with a decrease in aging and other chronic disorders [93]. Activating SIRT1 has been shown to lower oxidative stress, have anti-inflammatory effects, and improve mitochondrial function, according to earlier reports [94,95,96]. By restricting mTOR phosphorylation (ser 2884), AMPK activation aids autophagy induction and prevents depolarized and ROS-producing mitochondria while concurrently promoting the production of new functional mitochondria in peripheral nerves under hyperglycaemic conditions [97,98]. Through the stimulation of FOXO3a, SIRT1 and AMPK activation have also been shown to have the capacity to reverse metabolic memory by reversing the epigenetic modifications caused by high glucose in stem cells. Previously, it was observed that SIRT1 expression decreased oxidative stress in disease-related experimental models via deacetylating FOXO3a activity [99,100]. By upregulating genes involved in the antioxidant defense, FOXO3a reduces diabetic neuropathic pain. Deacetylation of FOXO3a by SIRT1 decreased oxidative stress in the N2A cells and helped to alleviate mitochondrial dysfunction, which helped to manage diabetic neuropathy, according to another study performed on STZ-induced diabetic mice [101]. In our studies, increased expression of NFκB and COX-2 in the sciatic nerves of diabetic SD rats and increased the expression levels of NFκB and COX-2 in Schwann cells exposed to high glucose, which provided evidence of the neuroinflammation caused by hyperglycemia. Both in the STZ-induced diabetic rats and in the Schwann cells exposed to hyperglycemia, AMPK activation by BC and CBD decreased the neuroinflammation linked to NFκB signaling, which is consistent with past studies on AMPK-directed NFκB suppression [74,102].

The capacity of BC and CBD to reverse the impairments of nerve blood flow in diabetic rats may be partly due to the reduction in oxidative damage to the peripheral nerves. Through involvement in the Nrf2/NFκB crosstalk, the BC and CBD combination likely affected cellular inflammation, the redox state, and the response to oxidative stress [41,103]. Our hypothesis that the BC and CBD combination mediated neuroprotection in DN due to their antioxidant and anti-inflammatory activity is supported by increased sirT3, SOD2, and Nrf2 in the diabetic rats treated with the BC and CBD combination and decreased NLRP3, NFκB, and COX2 levels, which is also supported by other prior findings [104,105,106].

Activation of NLRP3 has been linked to a variety of diseases, from cancer to infectious disorders [107,108]. NLRP3’s role in diabetic neuropathy has yet to be fully determined. Even though NLRP3 is the inflammasome that has been most thoroughly studied, the mechanisms regulating its targets are not completely elucidated. Discovering and developing treatment candidates that target NLRP3 in many inflammatory diseases, including DN, will depend on understanding the cellular and molecular features of NLRP3 inflammasome activation and inhibition. Some studies have recently clarified the potential function of autophagy and mitochondrial quality control in the regulation of NLRP3 activation [109,110]. Evidence is mounting that NLRP3 activation is crucial to the development of diabetic neuropathy after nerve damage [111,112]. Thus, elevating Nrf2 activity may induce autophagy, which may lessen NLRP3-induced neuroimmune abnormalities while also enhancing the antioxidant status of the neurons.

Under typical conditions, Nrf2 is sequestered in the cytoplasm via the Kelch-like ECH-associated protein 1 (Keap1)-Nrf2 complex, where it is susceptible to ubiquitin-mediated destruction. Oxidative stress promotes Keap1–Nrf2 complex separation. Next, Nrf2 enters the nucleus and activates ARE to regulate the transcription of antioxidant genes. The primary regulator that is involved in the endogenous antioxidant defense system is Nrf2 [95,113,114,115,116]. In rats with STZ-induced diabetes, the loss of the Nrf2-mediated antioxidant defense due to T1DM increased the oxidative burden, which has previously been linked to nerve damage [117]. We also found decreased Nrf2 signaling and the expression of the enzyme NQO1 under diabetes circumstances, further corroborating the earlier findings [118,119]. Diabetic rats treated with the BC and CBD combination had significantly (*p* < 0.001) increased Nrf2 antioxidant response element transcription and promoted the production of antioxidant enzymes such NQO1 and HO-1. In addition to the in vivo evidence, it was observed that the BC and CBD combination significantly (*p* < 0.001) increased Nrf2-mediated signaling in Schwann cells that had been damaged by HG [120]. In order to prevent the protein aggregates and damaged cellular components in neurological diseases, autophagy serves as a neuroprotective strategy [121,122]. The decreased expression of LC3B and Beclin1 that we observed in our studies underscores that the production of autophagosomes was inhibited in hyperglycemic circumstances. Beclin1 aids in the creation of the phagophore and the recruitment of other autophagy-related proteins like LC3B to the phagophore, which results in the formation of autophagosomes with the aid of the class III PI3K enzyme Vsp34 [123]. Treatment with the BC and CBD combination significantly (*p* < 0.001) increased the cleavage of LC3B from LC3B-I and the expression of Beclin1, which is a sign that autophagy was increased. By clearing away the damaged mitochondria, this enhanced autophagosome production further improved mitochondrial function due to the BC and CBD treatment [32,84,120].

Diabetes and the accompanying hyperglycemia have varied effects on different organ metabolic processes. When a damage sensing mechanism is activated, PINK1, a resident protein of the inner membrane of mitochondria (IMM), begins to build up at the outer mitochondrial membrane (OMM), allowing the removal of unhealthy and dysfunctional mitochondria. In our studies, we observed that the BC and CBD combination exhibited a significant potential role in autophagy by increasing the PINK1 and PARKIN levels [84,124]. Overall, the results of our current studies have successfully demonstrated the neuroprotective potential of the BC and CBD combination in HG-induced neurotoxicity and experimental hyperglycemia in rats. By balancing the two opposed processes of mitochondrial biogenesis and autophagy, SIRT1, AMPK, and Nrf2 activation by BC and CBD results in effective metabolic sensing related to mitochondrial function (Figure 13). To provide the required bioenergetic needs for enhanced impulse conduction and neurotransmission, the mitochondrial function must be optimized. Additionally, by making nociceptors more sensitive, activating SIRT1 and AMPK lowers oxidative damage, neuroinflammation, and nociceptive firing brought on by inflammatory cytokines. Because SIRT1 and AMPK have a protective impact against the neurodegenerative alterations associated with DN through modifications in mitochondrial quality control, they can be taken into consideration while creating cutting-edge strategies for the management of DN.

## 5. Conclusions

In summary, the present studies demonstrated that STZ- and HG-induced oxidative and nitrosative stress play a crucial role in the pathogenesis of diabetic neuropathy. The functional, behavioral, and molecular deficits were due to oxidant-induced damage, neuroinflammation, and bioenergetic deficits. These pathological consequences of nerve injury have been attenuated by the combination of CBD and BC in vitro and in vivo (Figure 13). Our findings suggest that the enhanced neuroprotective effects of combination therapy may be attributable to simultaneous inhibition of oxidative stress, neuroinflammation, and NLRP3, as well as activation of Nrf2. Hence, the combination therapy could be suggested as a potential strategy that can be further pursued for the management of STZ- and HG-induced diabetic neuropathy.

## Figures and Tables

**Figure 1 biomedicines-12-01442-f001:**
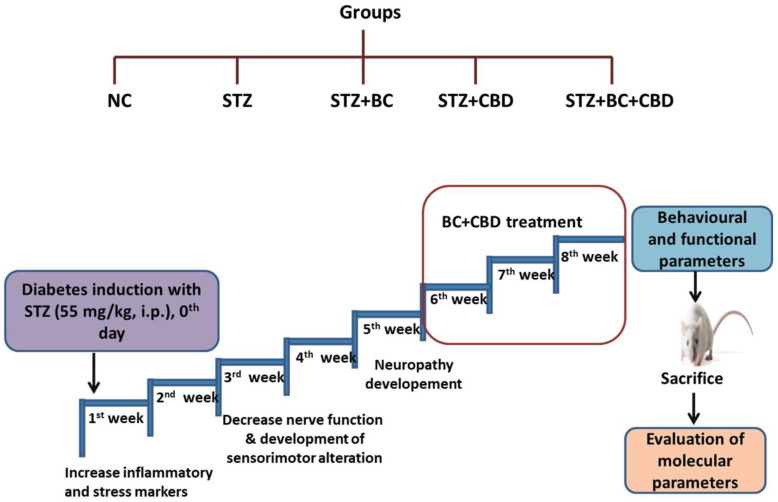
Experimental design. There were 5 groups of healthy male SD rats. A single dose of STZ, 55 mg/kg, i.p. was given to the SD rats and BC (30 mg/kg, i.p), CBD (15 mg/kg, i.p.), and BC+CBD (30 mg/kg, i.p, and 15 mg/kg, i.p., respectively) were given for the last 3 weeks. Functional and behavioural parameters were recorded for normal control SD rats (NC, *n* = 6), STZ-induced SD rats (STZ, *n* = 8), STZ-induced SD rats with BC (STZ+BC, *n* = 6), STZ-induced SD rats with CBD (STZ+CBD, *n* = 6), and STZ-induced SD rats with BC and CBD (STZ+BC+CBD, *n* = 6).

**Figure 2 biomedicines-12-01442-f002:**
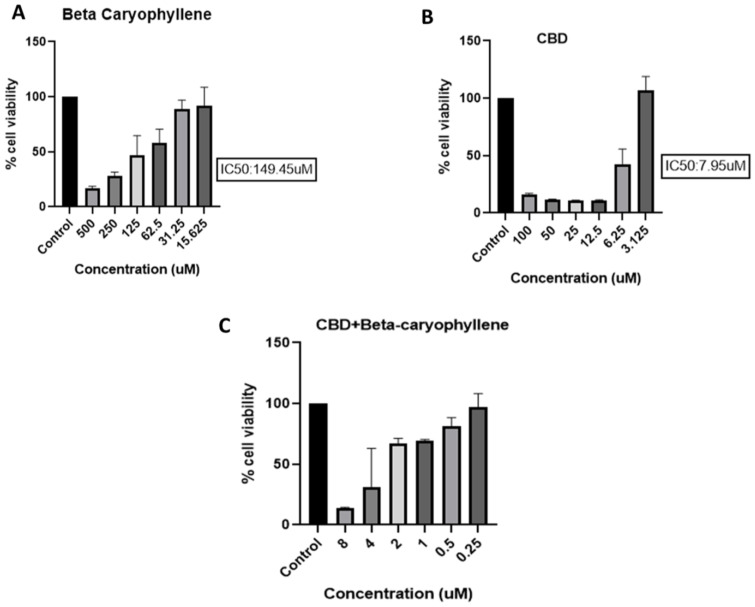
Effects of BC, CBD, and their combination on cell viability. BC and CBD have shown significant toxicity at above 200 µM and 20 µM concentrations, respectively. Schwann cells were treated at different concentrations of (**A**) BC (500–15.625 µM), (**B**) CBD (100–3.125 µM), and (**C**) BC+CBD (8–0.25 µM), respectively. Data values are represented as means ± SEM (*n* = 3).

**Figure 3 biomedicines-12-01442-f003:**
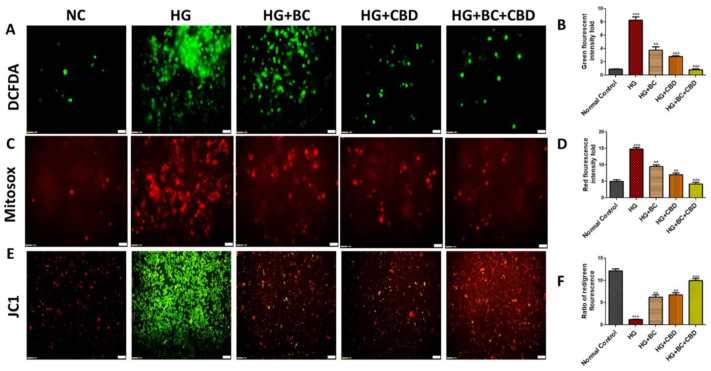
Effects of BC, CBD, and their combination on ROS, superoxide, and mitochondrial membrane potential in HG-induced Schwann cells. (**A**,**B**) Fluorescence microscopic images and bar graph of Schwann cells indicating the reactive oxygen species. (**C**,**D**) Fluorescence microscopic images and bar graph of Schwann cells indicating mitochondrial membrane potential. (**E**,**F**) Fluorescence microscopic images and bar graph of Schwann cells indicating mitochondrial superoxide generation. Photographs were captured at a 40× magnification. Scale shows a length of 50 µm. All the data values are represented as means ± SEM (*n* = 3). ^^^ *p* < 0.001, ^^ *p* < 0.01 vs. HG, *** *p* < 0.001 vs. Normal Control. Normal Control: normal Schwann cells; HG: Schwann cells exposed to high glucose (30 mM); HG+BC: HG-induced cells treated with BC 75 µM; HG+CBD: HG-induced cells treated with CBD 8 µM; HG+BC+CBD: High-glucose-induced cells treated BC and CBD with 75 µM and 3.64 µM, respectively.

**Figure 4 biomedicines-12-01442-f004:**
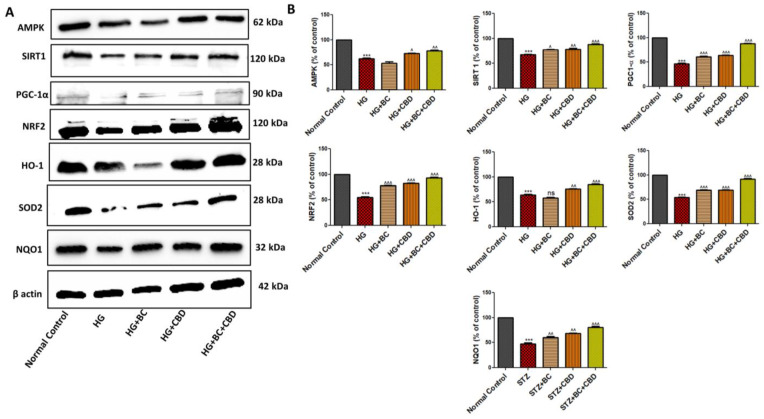
Effects of BC, CBD, and their combination on mitochondrial biogenesis and their antioxidant effects through the expression of AMPK, SIRT1, and NRF2 in HG-induced Schwann cells. (**A**) Western blots of respective proteins AMPK, SIRT1, NRF2, PGC-1α, HO1, SOD2, NQO1, and β-actin. (**B**) Densitometric analyses of the respective blots. All the data values are represented in the form of mean ± SEM (*n* = 3). ns = non significance, ^^^ *p* < 0.001, ^^ *p* < 0.01, ^ *p* < 0.05 vs. HG, *** *p* < 0.001 vs. Normal Control. Normal Control: normal Schwann cells; HG: Schwann cells exposed to high glucose (30 mM); HG+BC: HG-induced cells treated with BC 75 µM; HG+CBD: HG-induced cells treated with CBD 8 µM; HG+BC+CBD: high-glucose-induced cells treated with BC and CBD with 75 µM and 3.64 µM, respectively.

**Figure 5 biomedicines-12-01442-f005:**
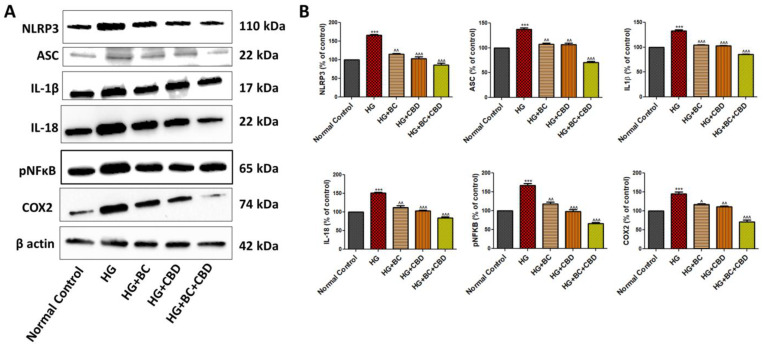
Effects of BC, CBD, and their combination on mitochondrial function and autophagy in HG-induced Schwann cells. (**A**) the Western blotting of NLRP3, ASC, IL1β, IL18, pNFkB, COX2, and β-actin. (**B**) Densitometric analysis of NLRP3, ASC, IL-1β, IL18, pNFkB, and COX2 expression in Schwann cells. All data values are expressed as means ± SEM (*n* = 3). ^^^ *p* < 0.001, ^^ *p* < 0.01, ^ *p* < 0.05 vs. HG, *** *p* < 0.001, vs. Normal Control. Normal Control: normal Schwann cells; HG: Schwann cells exposed to high glucose (30 mM); HG+BC: HG-induced cells treated with BC 75 µM; HG+CBD: HG-induced cells treated with CBD 8 µM; HG+BC+CBD: high-glucose-induced cells treated BC and CBD with 75 µM and 3.64 µM, respectively.

**Figure 6 biomedicines-12-01442-f006:**
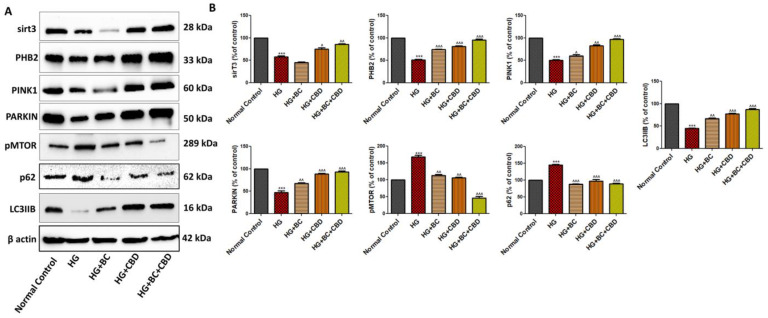
Effects of BC, CBD, and their combination on neuroinflammation in HG-induced Schwann cells. (**A**) Western blotting of sirT3, PHB2, PARKIN, PINK1, pMTOR, p62, LC3IIB, and β-actin (**B**) Densitometric analysis of sirT3, PHB2, PARKIN, PINK1, pMTOR, p62, and LC3IIB levels in Schwann cells. All data values are presented as means ± SEM (*n* = 3). ^^^ *p* < 0.001, ^^ *p* < 0.01, ^ *p* < 0.05, vs. HG, *** *p* < 0.001, vs. Normal Control. Normal Control: normal Schwann cells; HG: Schwann cells exposed to high glucose (30 mM); HG+BC: HG-induced cells treated with BC 75 µM; HG+CBD: HG-induced cells treated with CBD 8 µM; HG+BC+CBD: high-glucose-induced cells treated with BC and CBD with 75 µM and 3.64 µM, respectively.

**Figure 7 biomedicines-12-01442-f007:**
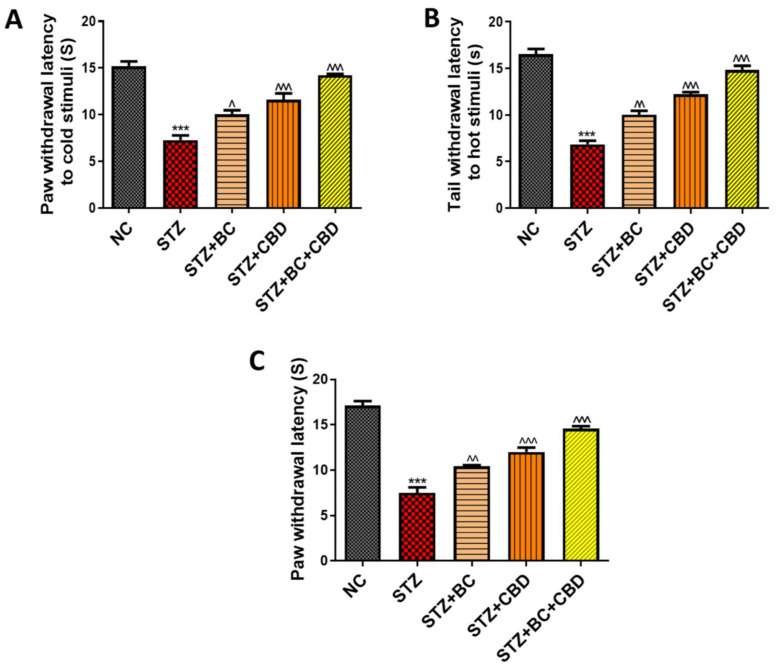
Effects of BC, CBD, and their combination on thermal stimuli in diabetic rats. The thermal stimuli (**A**) cold immersion test (**B**) hot immersion test, and (**C**) Hargreaves test, (*n* = 6). All the data values are presented as means ± SEM. ^^^ *p* < 0.001, ^^ *p* < 0.01, ^ *p* < 0.05 vs. STZ, *** *p* < 0.001 vs. NC. NC: normal control; STZ: diabetic control, STZ+BC; diabetic control treated with BC 30 mg/kg, STZ+CBD; diabetic control treated with CBD 15 mg/kg; STZ+BC+CBD: diabetic control treated with BC and CBD at 30 mg/kg and 15 mg/kg body wt, respectively.

**Figure 8 biomedicines-12-01442-f008:**
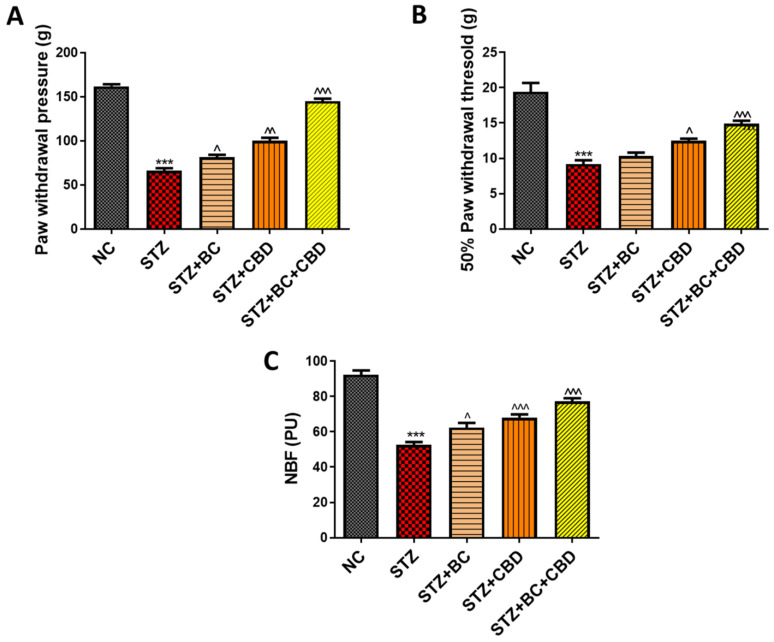
Effects of BC, CBD, and their combination on mechanical stimuli and nerve function in diabetic rats. The mechanical stimuli, (**A**) Von Frey filaments test, (**B**) Randall–Selitto apparatus test (*n* = 6), (**C**) nerve blood flow test (NBF) (*n* = 3). All the data values are presented as means ± SEM. ^^^ *p* < 0.001, ^^ *p* < 0.01, ^ *p* < 0.05 vs. STZ, *** *p* < 0.001 vs. NC. NC: normal control; STZ: diabetic control; STZ+BC: diabetic control treated with BC 30 mg/kg; STZ+CBD: diabetic control treated with CBD 15 mg/kg; STZ+BC+CBD: diabetic control treated with BC and CBD at 30 mg/kg and 15 mg/kg body wt, respectively.

**Figure 9 biomedicines-12-01442-f009:**
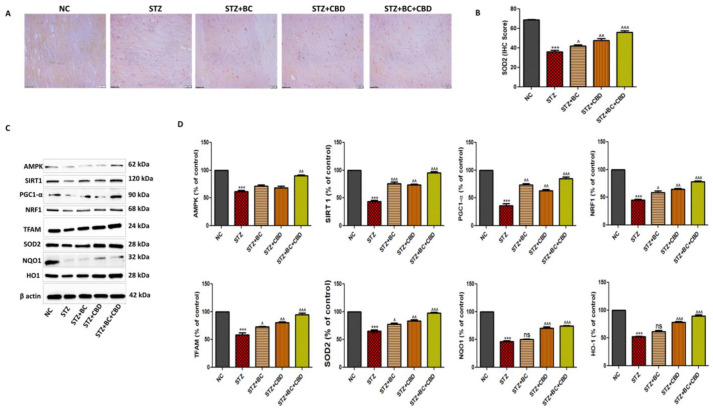
Effects of BC, CBD, and their combination on AMPK, SIRT1-mediated mitochondrial biogenesis in SD rats. (**A**) The expression of SOD2 immunopositivity in the SD rats’ sciatic nerves. (**B**) The bar graph of the respective images. Photographs were taken at 40× magnification. Scale bars show a length of 50 µm. (**C**) Western blots of AMPK, SIRT1, NRF1, PGC-1α, TFAM, NQO1, SOD2, HO1 and β-actin in the sciatic nerve of the diabetic rats. (**D**) The corresponding graphical representation of the blots based on the densitometric analysis. All the data values are represented as means ± SEM (*n* = 3). ns = non significance ^^^ *p* < 0.001, ^^ *p* < 0.01, ^ *p* < 0.05 vs. STZ, *** *p* < 0.001, vs. NC. NC: normal control; STZ: diabetic control; STZ+BC: diabetic control treated with BC 30 mg/kg; STZ+CBD: diabetic control treated with CBD 15 mg/kg; STZ+BC+CBD: diabetic control treated with BC and CBD at 30 mg/kg and 15 mg/kg body wt, respectively.

**Figure 10 biomedicines-12-01442-f010:**
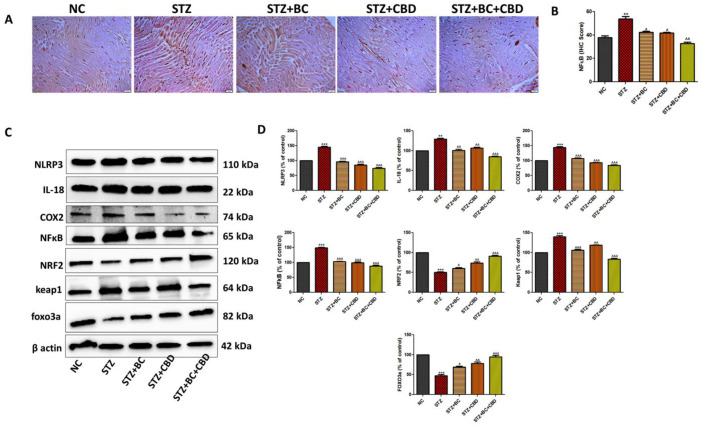
Effects of BC, CBD, and their combination on inflammasome and Nrf2-linked antioxidant defense in SD rats. (**A**) Representative immunohistochemical representation of NFkB immunopositivity in the sciatic nerves of the rats. (**B**) Bar graph of the respective images. Photographs were captured at 40× magnification. Scale bars show a length of 50 µm. (**C**) Representative Western blots of NLRP3, IL-18, COX2, NFkB, Nrf2, keap1, Foxo3a, and β-actin in the diabetic rats. (**D**) Corresponding graphical representation of the blots based on a densitometric analysis. All the data values are represented as means ± SEM (*n* = 3). ^^^ *p* < 0.001, ^^ *p* < 0.01, ^ *p* < 0.05 vs. STZ, *** *p* < 0.001, *** p* < 0.01, vs. NC. NC: normal control; STZ: diabetic control; STZ+BC: diabetic control treated with BC 30 mg/kg; STZ+CBD: diabetic control treated with CBD 15 mg/kg; and STZ+BC+CBD: diabetic control treated with BC and CBD at 30 mg/kg and 15 mg/kg body wt, respectively.

**Figure 11 biomedicines-12-01442-f011:**
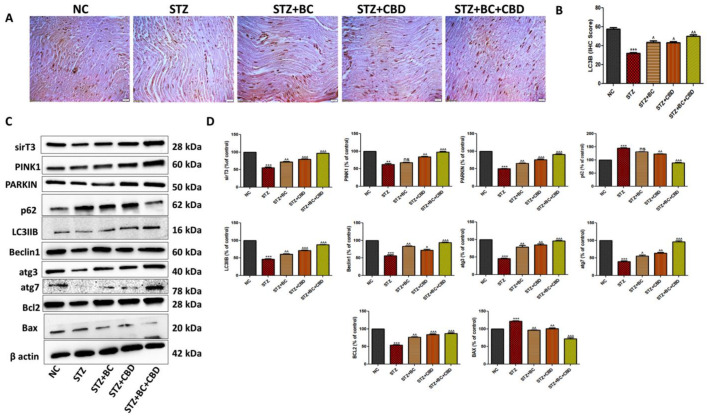
Effects of BC, CBD, and their combination on autophagy in SD rats. (**A**) The expression of LC3B immunopositivity in the sciatic nerves of rats. (**B**) Bar graph for the respective photographs. Photographs were taken at 40× magnification. Scale bars show a length of 50 µm. (**C**) Representative Western blots of SirT3, PINK1, PARKIN, p62, LC3B, pMTOR, atg3, atg7, BCL2, and β-actin in diabetic rats. (**D**) Corresponding graphical representation of the blots based on a densitometric analysis. All the data values are represented as means ± SEM (*n* = 3). ns = non significance, ^^^ *p* < 0.001, ^^ *p* < 0.01, ^ *p* < 0.05 vs. STZ, *** *p* < 0.001, ** *p* < 0.01 vs. NC. NC: normal control; STZ: diabetic control; STZ+BC: diabetic control treated with BC 30 mg/kg; STZ+CBD: diabetic control treated with CBD 15 mg/kg; STZ+BC+CBD: diabetic control treated with BC and CBD at 30 mg/kg and 15 mg/kg body wt, respectively.

**Figure 12 biomedicines-12-01442-f012:**
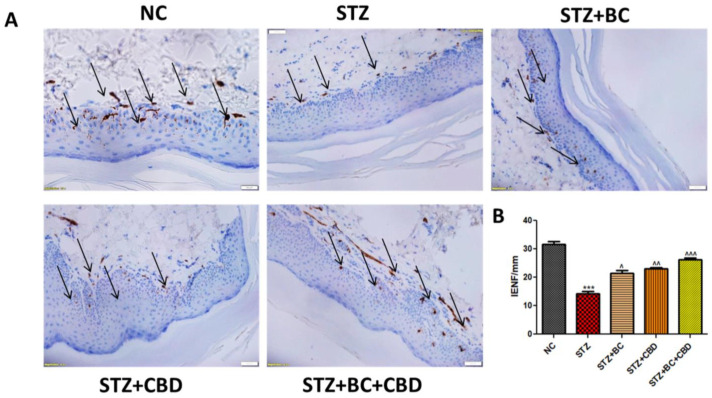
Effects of BC, CBD, and their combination on IENF degeneration in SD rats. (**A**) Representative immunohistochemistry images of PGP 9.5 in the diabetic rat planter skin. The photographs were captured at 40× magnification. Scale bars show a length of 50 µm. (**B**) Bar graph shows IENF/mm. The results are represented as means ± SEM (*n* = 3). ^^^ *p* < 0.001, ^^ *p* < 0.01, ^ *p* < 0.05 vs. STZ, *** *p* < 0.001, vs. NC. NC: normal control; STZ: diabetic control; STZ+BC: diabetic control treated with BC 30 mg/kg; STZ+CBD: diabetic control treated with CBD 15 mg/kg; and STZ+BC+CBD: diabetic control treated with BC and CBD at 30 mg/kg and 15 mg/kg body wt, respectively.

**Figure 13 biomedicines-12-01442-f013:**
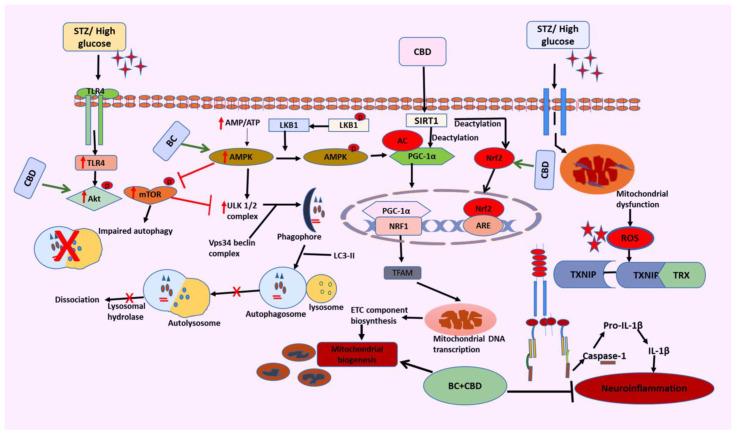
Hypothesized mechanism of BC and CBD via AMPK/SIRT1/Nrf2 activation and NLRP3 inactivation. Peripheral nerves may accumulate more glucose and ROS because of diabetes-associated oxidative stress/nitrosative stress and dysfunctional mitochondria, which act as priming and activation signals for the activation of NLRP3 and release of IL-1β. On the other side, NLRP3 activation comes from abnormal clearance of damaged mitochondria caused by defective autophagy, which fails to balance mitochondrial homeostasis. To combat the NLRP3-linked neuroinflammation that is thought to be responsible for the onset and maintenance of diabetic neuropathy, BC and CBD, which are antioxidants, enhance Nrf2-mediated p62 induction and improve both autophagy and antioxidant effects. On the other hand, activation of AMPK, SIRT1, and Nrf2 and the inhibition of mTOR by BC and CBD promote autophagy. PGC-1α deacetylation and associated transcriptional activation of NRF1-mediated mitochondriogenesis are both outcomes of BC and CBD, facilitating SIRT1. Additionally, ROS-linked neuroinflammation and oxidative stress can be prevented by activating AMPK and SIRT1. Under hyperglycemic conditions of DN, these AMPK and SIRT1-mediated impacts on redox balance and mitochondrial homeostasis can avoid neuronal damage. Overall, the ROS-directed mitochondrial dysfunction and oxidative stress within the cellular system are reduced by the newly synthesized healthy mitochondria and via autophagy. AMPK: AMP-activated protein kinase; SIRT1: silent mating type information-regulation 2 homolog; PGC-1α: peroxisome proliferator-activated receptor-γ coactivator; Nrf2: nuclear respiratory factor 2; p62: Sequestome1; TFAM: transcription factor A; mTOR: mammalian target of rapamycin.

## Data Availability

The data that support the findings of this study are available in the manuscript. In addition, data can be made available from the corresponding author upon reasonable request.

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
