# Peer review of "Cannabidiol and Beta-Caryophyllene Combination Attenuates Diabetic Neuropathy by Inhibiting NLRP3 Inflammasome/NFκB through the AMPK/sirT3/Nrf2 Axis"

_biomedicines, 2024, doi:10.3390/biomedicines12071442_

Round 1
Reviewer 1 Report
Comments and Suggestions for Authors
The study is well designed. Authors are requested to address following comments-
1. Authors should add make, model and country of instruments used in the study (e.g Von Frey Aesthesiometer, Randall Sellito etc.)
2. In figure 4 to 10, groups should be renamed as , normal control, diabetic control, diabetic+BC, diabetic+CBD and diabetic +BC+CBD
3. What was the basis for selection of doses of BC and CBD for the study?
4. What was the basis for selection of dose of combination of BC and CBD for the study?
5. Oxidative stress also plays important role in progression of diabetic neuropathy. Why these parameters were not assessed?
6. Authors can add detailed prevalence of diabetic neuropathy in various regions of the world.
7. Authors should add chemical structures of BC and CBD.
8. There are few approved treatments for diabetic neuropathy. Why authors have not kept a standard drug group in the present study?
Author Response
Reviewer #1: comments:
- Authors should add make, model and country of instruments used in the study (e.g Von Frey Aesthesiometer, Randall Sellito etc.)
Response: Authors have corrected it and mentioned it in the manuscript.
- In figure 4 to 10, groups should be renamed as, normal control, diabetic control, diabetic+BC, diabetic+CBD and diabetic +BC+CBD
Response: Authors have corrected the figure and the recommendations have been strictly followed
- What was the basis for selection of doses of BC and CBD for the study?
Response: The dose of BC and CBD were selected based on MTT assays from in vitro studies and correlating it to a in vivo dose. Furthermore, in vivo study dose was also selected based on literature review which matched well with our in vitro studies.
- What was the basis for selection of dose of combination of BC and CBD for the study?
Response: The combination dose was selected based on MTT assay which determined as to which ratio of the combination showed synergistic activity.
- Oxidative stress also plays important role in progression of diabetic neuropathy. Why these parameters were not assessed?
Response: Authors appreciated the reviewer suggestion and the authors have checked the oxidative markers with western blotting like SOD2.
- Authors can add detailed prevalence of diabetic neuropathy in various regions of the world.
Response: Authors have written about the prevalence of diabetic neuropathy in the world and the new additions reads like this: The number of people suffering from diabetes is anticipated to rise from the current 536.6 million cases to 783.2 million by 2045 [3]. Looking at the world demographics, 8.4% of Chinese, 48.1% of Sri Lankans, 29.2% of Souheast Asians, 56.2% of Yemenis, 39.5% of Jordanians, 71.1% Nigerians, 16.6% of Ghanaians, and 29.5% of Ethiopian have DPN [4]. Further, in Norh America 47percent of the patients with diabetes have some form of neuropathy.
- Authors should add chemical structures of BC and CBD.
Response: Authors have added the chemical structure.
- There are few approved treatments for diabetic neuropathy. Why authors have not kept a standard drug group in the present study?
Response: Thank you for your suggestion, however we have not used any standard because we were evaluating the mechanistic role of BC and CBD in DN. Further, there are very few drugs which work for DPN.
Reviewer 2 Report
Comments and Suggestions for Authors
I wanted to start by complimenting you on the work done both in vivo and in vitro. However, upon reading the paper, it will be necessary to improve some aspects of the data presentation.
1) In the INTRODUCTION:
-The presentation of the molecules that will be analyzed should be improved. For example, PGC-1a is mentioned in line 69 without context.
-CB1 and CB2 are mentioned in line 96, and CB2 is also mentioned in line 100. The description of these receptors and their activation in cells in general should be improved.
DOI: 10.3390/molecules26185576.
2) MATERIALS AND METHODS section:
- Experimental design:
--- It should be explained why two different concentrations of CBD are used when used alone or in combination with BC.
--- It is not indicated how the BC and CBD molecules are prepared, and whether the vehicles have been tested to show that their effect is similar to an untreated control condition. This should be done similarly to what was done in the In Vivo section.
- Cell viability determination: The MTT assay appropriately assesses viability and determines the optimal concentration to use. However, in the case of BC, a single concentration is used, whereas for CBD, two different concentrations are used.
- Before the ICH section, a section on the paraffin embedding methodology should be included.
- ICH: Specify that hematoxylin was used for nuclear staining.
3) In the RESULTS section:
- Lines 275-277 should be rephrased more clearly, separating the individual concentrations. For example, "BC, CBD respectively 75 and 8, and for BC+CBD respectively…."
- The described results need to be well elaborated in the text, not solely in Figure 2.
- In Figure 2, each histogram should be labeled as 2A, B, C, and so on, both in the text and in the caption.
- It's not clear whether the results of the experiments were obtained from a single, duplicated, or triplicated experiment. This should be added in the Figure 2 caption.
- In the caption of Figure 2, asterisks are indicated as significance markers but are not present atop the reference histograms.
- In Figure 3e and in the text, it would be clearer and easier to observe if letters C and D were moved to follow letters E and F, as DCFDA and MitoSOX are initially described, followed by JC1.
- The conditions HG-BC and HG-CBD are indicated in the histograms of Figure 3; therefore, their results should be explained at least once, along with which conditions they were compared to.
- When referring to JC1 in line 303, it should be clearly explained what the green and red colors indicate.
- In the caption of Figure 3, after "Schwann cells," a grammatical point should be inserted, as lines 293-294 seem to be a brief description and instead appear to continue from what is indicated by letters A and B.
- In the caption of Figure 3, the magnification used (40x) should not be indicated; instead, the length of the scale bar in the panel should be mentioned, a value that should also be included in at least one panel photo.
- In the histogram referring to HO-1 in Figure 4, the abbreviation "ns" is indicated for the condition HG+BC, but it's not specified in the caption.
- Caption 5 corresponds to Figure 6 and vice versa; please switch them accordingly.
- In line 363, "sirt3" is indicated differently from other instances, as is also the case next to the bands of the western blot.
- In line 374, next to "phospho-mTor," it should be indicated as "(pmTor)" as shown in Figure 6.
- In Figure 9A, the photo indicating STZ+CBD is larger compared to the others; resize it accordingly.
- The histogram in Figure 9B doesn't indicate the type of analysis performed to obtain the values from photo 9A.
- Both in the photo and caption, indicate the scale bar rather than magnification.
- Figures 4A, 5A, 9A, and 11A show the analysis of proteins with the same molecular weight (e.g., SOD 2 and HO1, both 28 kDa in Fig 9A), so the respective beta-actin loading control should be described in the caption. Otherwise, they cannot be normalized by the same beta-actin as the signal detection may be an artifact.
- For Figure 10B, indicate that the analysis was obtained from 10°.
- Also, for Figure 11 and related text, ensure consistency in the term "sirt3" throughout the work, as it's written in various ways.
- Like the other figures, indicate "SCALE BAR" instead of "40X" in Figure 11.
- In Figure 12A, the control condition should be labeled as NC, as consistent with the rest of the work.
- Specify in the caption of 12 the BAR.
- For 12B, specify the type of analysis used to obtain the histogram.
- Since arrows were used in Figure 12A, it should be explained in the text and caption what they indicate.
4) In the Discussion section:
- Provide a clearer discussion of the role of Foxo3a, as it seems only briefly mentioned.
- Delve deeper into the role of sirt1 at the neuronal level, considering its interaction with other antioxidant molecules as well. (DOI: 10.3390/nu13051418)
- Elaborate on the role of NLRP3 in a more comprehensive manner. (DOI: 10.1038/s41590-021-00886-5)
- In Figure 13, it's evident how CBD and STZ/HG act, but it's less clear how and where BC acts. BC appears in the figure only once, already bound within the cell, whereas the introduction mentions the role of the CB2 receptor with BC.
Author Response
Reviewer #2: comments:
1) In the INTRODUCTION:
-The presentation of the molecules that will be analyzed should be improved. For example, PGC-1a is mentioned in line 69 without context.
Response: Sirt1 deacetylates the PGC1a and that is why we mentioned it there and we have also done western blotting of PGC1a.
-CB1 and CB2 are mentioned in line 96, and CB2 is also mentioned in line 100. The description of these receptors and their activation in cells in general should be improved. DOI: 10.3390/molecules26185576.
Response: Thank you for the reviewer suggestion authors have added the references and the following section:
Cannabidiol pharmacology is intricate and is yet to be comprehensively investigated. It should be noted that CBD has a low affinity for cannabinoid receptors CB1 and CB2 and is thought to act through other targets (e.g. 5-Hydroxytrypyamine (5HT1A) or TRPV1 pain channels [49]. β-caryophyllene (BC), a sesquiterpene which is additionally present in clove & black pepper, is one of the terpenes that are also most prevalent in cannabis. It has been shown by various investigators that BC is a naturally occurring, selective CB2 receptor agonist with such positive effects that include analgesia, antioxidant protection, anti-inflammation, and neuroprotection [50,51]. Therefore, there is a need to determine the beneficial and undesirable (negative) consequences of the medical use of cannabinoids given the fast-changing legislation affecting access to cannabinoids and promote interest in the potential medical applications of cannabis (especially for DN) [52,53].
2) MATERIALS AND METHODS section:
- Experimental design: --- It should be explained why two different concentrations of CBD are used when used alone or in combination with BC.
Response: we have used two concentrations of CBD to show combination effect of BC and CBD. The main goal was to show some dose response effect also.
Comment: It is not indicated how the BC and CBD molecules are prepared, and whether the vehicles have been tested to show that their effect is similar to an untreated control condition. This should be done similarly to what was done in the In Vivo section.
Response: Drugs were dissolved in the vehicles which is mentioned in the methods section. We have done many studies with the vehicle with Schwann cells, and it has no toxicity effects.
Both, CBD and BC were dissolved in a vehicle comprised of 5% DMSO, 5% ethanol, 5% Tween 80, and 85% saline [83].
Comment: Cell viability determination: The MTT assay appropriately assesses viability and determines the optimal concentration to use. However, in the case of BC, a single concentration is used, whereas for CBD, two different concentrations are used.
Response: Based on MTT assay we have decided the dose of BC and CBD combination where we kept BC concentration constant while CBD concentration was varied.
Comment: Before the ICH section, a section on the paraffin embedding methodology should be included. - ICH: Specify that hematoxylin was used for nuclear staining.
Response: Authors have added the hematoxylin nuclear staining in the text. The paragraph now reads like his: The sections were counterstained with hematoxylin, dehydrated, and mounted with DPX. To ascertain the immunopositivity, each segment was examined with OLYMPUS IX73 inverted microscope. Image analysis were done by image J software [84].
Comment: In the RESULTS section: Lines 275-277 should be rephrased more clearly, separating the individual concentrations. For example, "BC, CBD respectively 75 and 8, and for BC+CBD respectively…."
Response: Authors have corrected the manuscript result part.
BC, CBD, and their combination effect on the cell viability MTT results revealed that the dose of BC, CBD & BC+CBD at 75 µM, 8 µM & 3.65 µM, respectively, was suitable to evaluate their protective potential in high glucose exposed Schwann cells (Fig 2).
Comment: The described results need to be well elaborated in the text, not solely in Figure
Response: Authors have corrected the manuscript as follows. BC and CBD showed significant toxicity at above 200 µM and 20 µM concentrations respectively. Schwann cells were given treatment at different concentrations of (2A) BC (500-15.625 µM), (2B) CBD (100-3.125 µM) & (2C) BC+CBD (8-0.25 µM) respectively. Data values are represented as mean±SEM (n=3).
Comment: In Figure 2, each histogram should be labeled as 2A, B, C, and so on, both in the text and in the caption.
Response: Authors have added the figure number in both figure and text.
Comment: It's not clear whether the results of the experiments were obtained from a single, duplicated, or triplicated experiment. This should be added in the Figure 2 caption.
Response: Authors have done n=3 and now his has been added in the figure caption.
Comment: In the caption of Figure 2, asterisks are indicated as significance markers but are not present atop the reference histograms.
Response: Authors have removed the asterisks from the text.
Comment: In Figure 3e and in the text, it would be clearer and easier to observe if letters C and D were moved to follow letters E and F, as DCFDA and MitoSOX are initially described, followed by JC1.
Response: Authors have corrected the figure as per reviewer suggestion.
Comment: The conditions HG-BC and HG-CBD are indicated in the histograms of Figure 3; therefore, their results should be explained at least once, along with which conditions they were compared to.
Response: The authors have added in the manuscript result section. It now reads as follows:
BC and CBD treatments at 75 and 3.65 µM, respectively, resulted in a significant (p < 0.001) reduction in the ROS and superoxide levels as compared to HG condition group (Fig 3A & 3D).
Comment: When referring to JC1 in line 303, it should be clearly explained what the green and red colors indicate.
Response: Authors have addressed his concern in the manuscript result section. Now it reads as follows:
JC1 labelling showed two separate cellular populations, a green fluorescence that indicates JC1 in its monomeric state and an orange fluorescence that signals the formation of JC1 aggregates which means negatively polarized mitochondria.
Comment: In the caption of Figure 3, after "Schwann cells," a grammatical point should be inserted, as lines 293-294 seem to be a brief description and instead appear to continue from what is indicated by letters A and B.
Response: Authors have corrected the sentences.
Comment: In the caption of Figure 3, the magnification used (40x) should not be indicated; instead, the length of the scale bar in the panel should be mentioned, a value that should also be included in at least one panel photo.
Response: Authors have mentioned in the figure legend.
Photographs were captured at 40x magnification. Scale length represents 50 µm
Comment: In the histogram referring to HO-1 in Figure 4, the abbreviation "ns" is indicated for the condition HG+BC, but it's not specified in the caption.
Response: Authors have mentioned ns in the manuscript figure legend.
Comment: Caption 5 corresponds to Figure 6 and vice versa; please switch them accordingly.
Response: Authors have corrected the figure legend of figure 5 and figure 6.
Comment: In line 363, "sirt3" is indicated differently from other instances, as is also the case next to the bands of the western blot.
Response: Authors have corrected sirt3 with sirT3 in the manuscript.
Comment: In line 374, next to "phospho-mTor," it should be indicated as "(pmTor)" as shown in Figure 6.
Response: Authors have corrected the pMTOR in the manuscript.
Comment: In Figure 9A, the photo indicating STZ+CBD is larger compared to the others; resize it accordingly.
Response: Authors have corrected the size STZ+CBD in figure number 9A.
Comment: The histogram in Figure 9B doesn't indicate the type of analysis performed to obtain the values from photo 9A.
Response: The analysis was done by image J software, and it is mentioned in the caption.
Comment: Both in the photo and caption, indicate the scale bar rather than magnification.
Response: Authors have added the scale bar in the caption and photo.
Comment: Figures 4A, 5A, 9A, and 11A show the analysis of proteins with the same molecular weight (e.g., SOD 2 and HO1, both 28 kDa in Fig 9A), so the respective beta-actin loading control should be described in the caption. Otherwise, they cannot be normalized by the same beta-actin as the signal detection may be an artifact.
Response: Yes, we do agree with the reviewers, but we have taken different nitrocellulose membranes for the SOD2 and HO1 blots with the same sample.
Comment: For Figure 10B, indicate that the analysis was obtained from 10°.
Response: Figure 10B was analyzed by figure 10A with the image J software and now it is mentioned in the caption.
Comment: Also, for Figure 11 and related text, ensure consistency in the term "sirt3" throughout the work, as it's written in various ways.
Response: Authors have corrected sirT3 throughout the manuscript.
Comment: Like the other figures, indicate "SCALE BAR" instead of "40X" in Figure 11.
Response: Authors have indicated scale bar in figure 11 and also mentioned in the text.
Comment: In Figure 12A, the control condition should be labeled as NC, as consistent with the rest of the work.
Response: Authors have corrected the figure as Normal control (NC).
Comment: Specify in the caption of 12 the BAR.
Response: Authors have specified the caption in figure legend 12 and explained in the result section.
Comment: For 12B, specify the type of analysis used to obtain the histogram.
Response: Analysis was done by image J software, and this is now specified in the manuscript.
Comment: Since arrows were used in Figure 12A, it should be explained in the text and caption what they indicate.
Response: The arrows are meant to point to the nerve fiber. This has now been explained in the text.
Comment: In the Discussion section:
Provide a clearer discussion of the role of Foxo3a, as it seems only briefly mentioned.
Response: Authors have mentioned the role of foxo3a in the discussion part. It now reads as: By upregulating genes involved in antioxidant defense, FOXO3a reduces diabetic neuropathic pain. Deacetylation of FOXO3a by SIRT1 decreased oxidative stress in the N2A cells and helps to alleviate mitochondrial dysfunction, which assists in the management of diabetic neuropathy, which is also supported by another study performed wih STZ-induced diabetic mice [107].
Comment: Delve deeper into the role of sirt1 at the neuronal level, considering its interaction with other antioxidant molecules as well. (DOI: 10.3390/nu13051418)
Response: Authors have added references in the text and explained sirt1 role in DN. The new section reads as: The changes in cellular NAD+/NADH ratio are mostly caused by a metabolic protein known as Sirtuin-1 [8,9]. By deacetylating transcription factors, Sirtuin-1 (SIRT1) stimulates downstream targets involved in biogenesis of mitochondria and antioxidant défense [10,11]. SIRT1 signaling is essential for DN because it regulates mitochondrial function and antioxidant enzymes [11–13]. Numerous studies have demonstrated that SIRT1 suppresses inflammatory responses by inhibiting the NLRP3 inflammasome in vascular endothelial cells [19,20]. Multiple investigations have also shown that SIRT1 activation is neuroprotective in DN in part by enhancing mitochondrial bioenergetics and autophagy [21,22]. Additional studies have also demonstrated that AMPK and SIRT1 help to reduce DN by improving mitochondrial function through the PGC-1α and Nrf2 axis [23–25].
Comment: Elaborate on the role of NLRP3 in a more comprehensive manner. (DOI: 10.1038/s41590-021-00886-5)
Response: authors have added the reference in the text and explained NLRP3 inflammasome in the manuscript. The new addition is as follows: Activation of NLRP3 has been linked to a variety of diseases from cancer to infectious disorders, [114,115]. NLRP3 role in diabetic neuropathy have yet to be fully determined. Even though NLRP3 is the inflammasome that has been most thoroughly studied, mechanisms regulating its targets are not completely elucidated. Discovering and developing treatment candidates that target NLRP3 in many inflammatory disease situations, including DN, therefore will depend on understanding the cellular and molecular features of NLRP3 inflammasome activation and inhibition. Recent studies have clarified the potential function of autophagy and mitochondrial quality control in the regulation of NLRP3 activation [116,117]. Evidence is mounting that NLRP3 activation is crucial to the development of diabetic neuropathy after nerve damage [118,119]. Thus, elevating Nrf2 activity may induce autophagy, which may lessen NLRP3-induced neuroimmune abnormalities while also enhancing the antioxidant status of the neurons.
Comment: In Figure 13, it's evident how CBD and STZ/HG act, but it's less clear how and where BC acts. BC appears in the figure only once, already bound within the cell, whereas the introduction mentions the role of the CB2 receptor with BC.
Response: BC has role in CB2 receptor as analgesic and it also acts as antioxidant, anti-inflammatory and neuroprotection in action. β-caryophyllene (BC), a sesquiterpene which is additionally present in clove & black pepper, is one of the terpenes that are also most prevalent in cannabis. It has been demonstrated that BC is a naturally occurring, selective CB2 receptor agonist with therapeutic effects that include analgesia, antioxidant protection, anti-inflammation, and neuroprotection [50,51]. Therefore, there is a need to determine the beneficial and undesirable (negative) consequences of the medical use of cannabinoids given the fast-changing legislation affecting access to cannabinoids and promote interest in the potential medical applications of cannabis (especially for DN) [52,53].
- Blanton, L. Yin, J. Duong, K. Benamar, Cannabidiol and Beta-Caryophyllene in Combination: A Therapeutic Functional Interaction, Int. J. Mol. Sci. 23 (2022). https://doi.org/10.3390/IJMS232415470.